# Antibiotic Susceptibility Patterns and Virulence-Associated Factors of Vancomycin-Resistant Enterococcal Isolates from Tertiary Care Hospitals

**DOI:** 10.3390/antibiotics12060981

**Published:** 2023-05-29

**Authors:** Arockia Doss Susai backiam, Senbagam Duraisamy, Palaniyandi Karuppaiya, Senthilkumar Balakrishnan, Balaji Chandrasekaran, Anbarasu Kumarasamy, Amutha Raju

**Affiliations:** 1Department of Microbiology, Vivekanandha College of Arts and Science for Women (Autonomous), Tiruchengode 637303, India; 2Department of Marine Biotechnology, Bharathidasan University, Tiruchirappalli 620024, Indiaanbumbt@bdu.ac.in (A.K.); 3Division of Biological Sciences, Tamil Nadu State Council for Science and Technology, Chennai 600025, India; nbsenthilkumar@gmail.com; 4Irma Lerma Rangel School of Pharmacy, Texas A & M University, Kingsville, TX 77843, USA; biobalajic@gmail.com; 5Department of Biotechnology, Periyar University Centre for Post Graduate and Research Studies, Dharmapuri 635205, India

**Keywords:** antibiotic resistance, vancomycin-resistant enterococcus, virulence factors, multi-drug resistance, gelatinase, protease, hemolysis, biofilm

## Abstract

This study explored the prevalence of multi-drug resistance and virulence factors of enterococcal isolates obtained from various clinical specimens (*n* = 1575) including urine, blood, pus, tissue, catheter, vaginal wash, semen, and endotracheal secretions. Out of 862 enterococcal isolates, 388 (45%), 246 (29%), 120 (14%), and 108 (13%) were identified as *Enterococcus faecalis*, *Enterococcus faecium*, *Enterococcus durans*, and *Enterococcus hirae*, respectively, using standard morphological and biochemical methods. The antibiotic resistance profile of all these enterococcal isolates was checked using the disc diffusion technique. High-level resistance was observed for benzylpenicillin (70%) and vancomycin (43%) among *E. faecalis* and *E. faecium* isolates, respectively. This study also revealed the prevalence of ‘multi-drug resistance (resistant to 3 antibiotic groups)’ among the vancomycin-resistant enterococcal strains, and this was about 11% (*n* = 91). The virulence determinants associated with vancomycin resistance (VR) were determined phenotypically and genotypically. About 70 and 39% of *E. faecalis* and *E. faecium* isolates showed to be positive for all four virulence factors (gelatinase, protease, hemolysin, and biofilm). Among the several virulence genes, *gelE* was the most common virulence gene with a prevalence rate of 76 and 69% among *E. faecalis* and *E. faecium* isolates, respectively. More than 50% of VRE isolates harbored other virulence genes, such *esp*, *asa*, *ace*, and *cylA*. Similarly, the majority of the VR enterococcal isolates (*n* = 88/91) harbored *vanA* gene and none of them harbored *vanB* gene. These results disclose the importance of VR *E. faecalis* and *E. faecium* and the associated virulence factors involved in the persistence of infections in clinical settings.

## 1. Introduction

Enterococci has been the most important causative agent of endocarditis and hospital-acquired infections for virtually from the last century. Enterococci are the most dominant pathogens among nosocomial urinary tract infection (UTI), bacteremia, and wound infections [1]. Nowadays, the majority of nosocomial infections are extremely difficult to treat with the available antibiotics, leading to high morbidity and mortality worldwide. Their intrinsic resistance to common antibiotics (penicillin, nalidixic acid, clindamycin, cephalosporin, and aminoglycoside) is the major reason for their survivability in a hospital environment [2]. Feasibly, their antibiotic resistance is acquired either via mutation or the horizontal transfer of genetic material. Glycopeptides, vancomycin, and teicoplanin are commonly used for treating Gram-positive bacterial infections, especially staphylococcal and enterococcal infections [3]. The widespread and frequent use of these glycopeptides in hospitals leads to the development of vancomycin-resistant enterococcus (VRE), which has a serious health and economic impact on healthcare professionals. Although the first case of VRE was reported in 1986 in the UK, recently, cases have been disseminated globally [2]. In the last two decades, VRE cases have increased twentyfold [4]. Nevertheless, in India, a smaller outbreak of VRE cases has been reported [2,5,6], though there has been a steady increase in enterococcal infection [7]. A study from North India reported that the prevalence of VRE and infection is slowly increased to 7.09%, and this increase leads to it being difficult to treat nosocomial infections [6]. The very limited number of studies on the prevalence of enterococci in the hospital environment, especially in Southern India, and the availability of poor data [5] proves the need for epidemiology studies of VRE cases in clinical areas.

Genetic and molecular analysis of VRE revealed that it may act as a reservoir of other antimicrobial-resistant genes. The major mechanism behind the enterococcal resistance to glycopeptide antibiotics includes changes in the peptidoglycan synthesis pathway, especially the D-alanine-D-alananine to either D-alanaine-D-lactate or D-alanaine-D-serine [8], and these changes result in glycopeptides resistance. Asymptomatic gut colonization by VRE leads to severe infections in immunocompromised patients, including patients under a prolonged period of antimicrobial treatment, human immunodeficiency virus (HIV)-infected patients, who are subject to a long stay in hospital, severely ill, etc. [9,10,11].

Enterococci are generally involved in causing endocarditis, urinary tract infections (UTIs), bacteremia, intra-abdominal and pelvic infection, skin, soft tissue, and wound infection [12]. The two most common pathogenic species of enterococci, *Enterococcus faecalis* and *Enterococcus faecium*, are capable of posing a public threat due to their antimicrobial resistance [13]. The adherence of *E. faecalis* to the host is the crucial step of pathogenesis. An aggregation substance (Agg), a surface protein that mediates the adherence of enterococcus to epithelial cells and the colonization of host tissue, is one of the virulence factors associated with adherence [14]. *E. faecalis* also secretes some other virulence factors that help to resist adverse conditions. Cytolysin (CylA) is another virulence factor that contributes to pathogenesis by lysing the blood cells [15]. Similarly, gelatinase and serine protease hydrolyzing gelatin and casein, respectively, play an important role in spreading the infection and also in biofilm formation [16]. Biofilm protects bacteria from the host defense and antimicrobials and confers their antimicrobial resistance to planktonic cells. Many of these factors (Agg, CylA, GelE, SprE, Esp, Hyl) determine the virulence of *Enterococcus* sp., and the genes that encode these virulence determinants [17] should be characterized for their proper diagnosis and management. Thus, the phenotypic and genotypic analysis of the virulence factors associated with antibiotic resistance is the one of the significant studies among VRE, especially *E. faecalis* and *E. faecium*, as they have now evolved as serious health problems. In this pipeline, the current study aims to phenotypically and genotypically characterize the associated virulence factors of *E. faecalis* and *E. faecium* isolates from various clinical specimens and their antibiotic susceptibility, with particular regard to vancomycin.

## 2. Results

In total, 1575 various samples including urine (*n* = 1121), blood (*n* = 234), pus (*n* = 86), tissue (*n* = 28), catheter (*n* = 6), vaginal wash (*n* = 32), semen (*n* = 48), and endotracheal secretion (*n* = 20) were collected during the period from 2018 to 2020. Among these, 77% of the samples were collected from hospitalized patients and 23% of the samples were taken from outpatients. The ratio of male and female of this study’s participants was 1:1. The complete details of various samples obtained from various patients are given in Table 1. From these various samples, a total of 974 enterococcal strains were isolated and analyzed in this study. In addition, other bacterial isolates were also detected, but these were left out from the study.

Among the 974 enterococcal isolates, well-grown 862 isolates were identified to the species level using standard microbiological identification methods. Among these 862 enterococcal isolates, 45% (*n* = 388) of isolates were *E. faecalis*, and 29% (*n* = 246) of isolates were *E. faecium*. Similarly, the remaining 27% of isolates were comprised of *E. durans* (*n* = 120) and *E. hirae* (*n* = 108). A high frequency of enterococcal strains was isolated from urine (39%), followed by pus (17%) and catheter (14%). The sociodemographic details of the patients and specimens used in this study are given in Table 2.

### 2.1. Antibiotic Susceptibility/Resistance Pattern

According to the antibiotic disk diffusion test (ADT), the highest frequency of resistance was observed for vancomycin (43%) and was followed by benzylpenicillin (37%) and erythromycin (35%) among the *E. faecalis* (*n* = 388) isolates. Similarly, in the case of *E. faecium* (*n* = 246) isolates, high-level resistance was noted for benzylpenicillin (70.3%) followed by gentamycin high level (58.9%), erythromycin (49.2%), teicoplanin (44.3%), and vancomycin (35%) (Figure 1). Similarly, a multi-drug resistance (MDR) phenotype was screened from the enterococcal isolates. Among the four species of *Enterococcus*, 22% (*n* = 54) of *E. faecium* isolates were resistant to five antibiotics (benzylpenicillin, gentamycin high level, erythromycin, teicoplanin, and vancomycin), followed by *E. faecalis* (*n* = 37, 10%). Likewise, 27 and 52% of *E. faecalis* and *E. faecium*, respectively, were found to be resistant to three antibiotics (benzylpenicillin, gentamycin at a high level, and erythromycin/vancomycin) (Table 3).

With respect to vancomycin resistance, 31% (*n* = 264/862) of VRE isolates belonging to three different species (*Enterococcus faecalis* (*n* = 167, 43%), *Enterococcus faecium* (*n* = 86, 35%) and *Enterococcus durans* (*n* = 11, 9%) were obtained from 1575 clinical specimens within the study period (2018–2020). Additionally, no more incidence of vancomycin resistance was observed amongst *E. hirae* strains.

The 91 enterococcal isolates (37 of *E. faecalis* and 54 of *E. faecium*) showing resistance to 5 antibiotics, namely benzylpenicillin, erythromycin, ciprofloxacin, teicoplanin, and vancomycin, were used to determine the MIC of these five antibiotics by the MDT (microdilution technique) and VITEK 2 system and compared to find out any discrepancy between these two results. According to the MDT, the MIC of benzylpenicillin for 72 VRE isolates was ≥32 µg/mL, and for the remaining 19 strains the MIC was 16 µg/mL. The VITEK 2 system also revealed a similar MIC value of benzylpenicillin. The MIC of erythromycin for 34 isolates was 4 µg/mL, and for the remaining 57 isolates the MIC value was >8 µg/mL. The same trend of the result was obtained from VITEK 2. The MDT revealed the MIC value of ciprofloxacin of 4 µg/mL (intermediate) for 7 VRE isolates, and for the remaining 84 isolates the MIC was 8 µg/mL (resistant region), whereas the VITEK 2 system displayed that the MIC of ciprofloxacin was 8 µg/mL (resistant region) for all the 91 isolates. Similarly, the MIC of teicoplanin was in the resistance region (MIC ≥ 32 µg/mL) for 88 isolates out of 91 MDR isolates. For the remaining three isolates, the MIC was found within a sensitive region (2 µg/mL), whereas according to the VITEK 2 system, the MIC of teicoplanin for 80 strains was ≥32 µg/mL, for 6 strains the MIC was 16 µg/mL (resistant region), and for the remaining 5 strains the MIC was 4 µg/mL (intermediate resistant). According to vancomycin, both the MDT and VITEK 2 system displayed an MIC of ≥32 µg/mL (resistant region) for all 91 strains (Table 4). Even though the MIC values acquired from the VITEK 2 system and reference method, MDT, were majorly agreed upon, some discrepancy errors were determined, and these are displayed in Table 5.

### 2.2. Detection of Virulence Phenotypes among VRE Isolates

The VR *E. faecalis* (*n* = 37) and *E. faecium* (*n* = 54) showing resistance to five antibiotics were evaluated for VR-associated virulence determinants such as gelatinase, protease, hemolysin, and biofilm production.

Appendix A show the complete details of the virulence phenotypic factors associated with VRE isolates. Gelatinase activity was detected in 71 VRE isolates (78%), and a clear zone was observed around the colonies. Among these 71 gelatinase-positive VRE, 31 (84%) were from *E. faecalis* and 40 (74%) were from *E. faecium*. Out of 59 isolates, 28 (76%) from *E. faecalis* and 31 (58%) from *E. faecium*) were positive for protease activity. According to the hemolysis assay, 33 (89%) *E. faecalis* isolates and 30 (56%) *E. faecium* strains were positive for hemolysis (Figure 2).

Biofilm detection was carried out using a crystal violet assay and showed that 32 (87%) *E*. *faecalis* and 36 (67%) *E. faecium* strains formed biofilm, but their biofilm-forming strength varied from the isolates. Among these, 51% (*n* = 19) of *E. faecalis* and 28% (*n* = 15) of *E. faecium* were found to be strong biofilm producers. Similarly, 27% (*n* = 10) of *E. faecalis* and 37% (*n* = 20) of *E. faecium* strains were moderate biofilm producers, and the remaining isolates (three of *E. faecalis* and seven of *E. faecium*) were weak producers (Figure 3). Overall, the results showed that out of the 37 isolates of *E. faecalis*, 26 (70%), and out of 54 isolates of *E. faecium*, 21 (39%) showed themselves to be positive for all of the phenotypic virulence factors tested in this study.

### 2.3. Distribution of Virulence Genes among the VRE Isolates

Genotypic screening for a set of seven virulence genes was performed among VRE isolates (*n* = 91). The virulence gene, *gelE*, was present with a high frequency (*E. faecalis* with 76% and *E. faecium* with 69%). Among the VR *E. faecalis*, the *gelE* was followed by *asaI* with 73%, *esp* with 70%, *ace* with 68%, *cylA* with 62%, and *hyl* with 5%. In the case of VR *E. faecium*, the *gelE* was followed by *esp* (63%), *asaI* (59%), *cylA*, *ace* (56%), *sprE* (48%), and *hyl* (30%) (Figure 2). According to the current results, the pattern of virulence genes among *E. faecalis* was *gelE*> *asaI* > *esp* > *ace* > *sprE* > *cylA* > *hyl*, and among the *E. faecium* the virulence genes pattern was *gelE* > *esp* > *asaI* > *cylA* and *ace* > *sprE* > *hyl* (Appendix A). Further, all the VR enterococcal isolates were genotypically assessed for their vancomycin resistance type (*vanA* and *vanB*). Out of 91 isolates, 33 and 50 from *E. faecalis* (89%) and *E. faecium* (93%) were found to carry the *vanA* gene, and none of the isolates carried the *vanB* gene.

### 2.4. Correlation between Virulence Phenotypes and Genotypes

The correlation between the phenotypic and genotypic virulence determinants of enterococcal isolates was statistically assessed at different levels against one another (chi-square and Fisher’s Exact test). The strongest association was found between the gelatinase activity and their encoding gene *gelE*. All the *gelE*-positive isolates of *E. faecalis* (*n* = 28) and *E. faecium* (*n* = 37) were gelatinase-positive and their correlation was highly significant (chi-square *p* = 0.0001, Fisher’s Exact test *p* = 0.0001) (Table 6). Among the 24 isolates of *sprE*-positive *E. faecalis*, 16 isolates showed protease activity, and the remaining 8 isolates showed negative protease activity (chi-square *p* = 0.0388, Fisher’s Exact test *p* = 0.0300). In the case of *E. faecium*, all the *sprE*-positive isolates ((*n* = 26) were positive for protease activity. The association between protease and *sprE* gene was highly significant (chi-square *p* = 0.0001, Fisher’s Exact test *p* = 0.0001) (Table 7). A similar statistically significant correlation was observed among hemolysis-positive isolates. Out of 23 *cylA*-positive *E. faecalis* isolates, all are positive for hemolysis activity (chi-square *p* = 0.0001, Fisher’s Exact test *p* = 0.0001). In the case of *E. faecium*, 28 *cylA*-positive isolates were positive for hemolysis and 2 *cylA*-positive isolates were negative for hemolysis (chi-square *p =* 0.0301, Fisher Exact test *p* = 0.0152) (Table 8)

The biofilm-producing ability of *E. faecalis* was correlated with *gelE* and *sprE*; however, their correlation was statistically nonsignificant (chi-square *p* = 0.9388, Fisher’s Exact test *p* = 0.3996 for *E. faecalis* and chi-square *p* = 0.7417, Fisher’s Exact test *p* = 0.6750 for *E. faecium*) (Table 9). Although, a good significant correlation was observed between *esp*, *ace*, and *asaI* genes and the biofilm-forming ability of *E. faecalis* and *E. faecium* isolates (Table 10).

## 3. Discussion

Nowadays, enterococci are considered one of the major nosocomial pathogens due to their intrinsic nature of resistance to several antibiotics including beta lactam and glycopeptide antibiotics. In particular, vancomycin-resistant enterococci (VRE) have become a challenging issue among public healthcare communities. Several studies have investigated the risk factors associated with VRE from various regions of India and few or no more studies have been reported from Tamil Nadu. The overall frequency of enterococcal isolates in this study was 55%. One earlier study [18] reported on the incidence of VRE at a tertiary care hospital in Northern India, and this frequency was about 7.9%. Another study [19] reported that the incidence of enterococcal infection in Ethiopia is about 6.2%. This high deviation might be due to variation in the study participants and a steady increase in the frequency of enterococcal infections among the hospital settings, etc.

The high frequency of enterococcal isolates was obtained from the age group of 41–65 (*n* = 359, 42%) and was followed by >65 age group (*n* = 202, 23%). The pediatric group (1–5 age group) showed the lowest frequency of enterococcal isolates (*n* = 46, 5%). Most of the enterococcal isolates were attained from the urology department (*n* = 329, 38%) and were followed by outpatients (281, 33%). A contrasting result was reported by Jahansepas et al. [17] who found that the highest frequency of enterococcal strains was from outpatient (29%) and internal ward (23%) rather than the urology department (8%) or other sources. According to the gender-wise distribution of enterococcal isolates, females showed a higher frequency (*n* = 514, 60%) than males (*n* = 348, 40%), and this may be due to the increased number of samples received and the high prevalence of urinary tract infections and another infection among women compared to men. This report is on the contrary to that of Karna et al. [1] who reported that enterococcal infection is more common in males (73.9%) than females. Among the various specimens collected, urine showed a high number of enterococcal isolates (*n* = 335, 39%). This result is correlated with Karna et al. [1] who reported that urine is the major source of enterococcal strains rather than any other sources such as blood, pus, etc.

The prevalence of VRE among the clinical specimens in this study was found to be 31%. Among them, 11% of VRE was obtained from outpatients and the remaining 20% of VRE was from hospitalized patients, including those in urology, ICU, surgery, pediatrics, and hematology departments. Similarly, the prevalence of VRE in Nepal was 25.3% [1] (Karna et al., 2019) and 21.4% in Iran [20]. However, the majority of previous studies [21,22,23] reported a lower prevalence of VRE compared to our study report. Such variation in VRE prevalence may be due to variation in geographical location, duration of hospital exposure, usage of medical devices, sample size, etc. [1]. The highest frequency of VR is observed among *E. faecalis* (43%) compared to that of *E. faecium* (35%) and *E. durans* (9.2%); however, multi-drug resistance (resistance to three antibiotics categories) is highly observed among *E. faecium* (22%) compared to that of *E. faecalis* (9.5%). This result is more prospectively comparable with other studies [1,24,25] which reported an increased occurrence rate of vancomycin resistance among *E. faecium* compared to *E. faecalis*. A nation-wide prevalence study on the epidemiology of VRE was carried out with 142 healthcare institutions in Switzerland, and it was disclosed that a high case of VRE infections may be due to nosocomial distribution [26]. Amongst the eleven antibiotics used in this study, the highest resistance of enterococcal isolates was observed to benzylpenicillin (38%) and vancomycin and erythromycin (31%), followed by high-level gentamycin (30%). This result is more in agreement with the report of Sreeja et al. [27] and Karna et al. [1]. Another study by Bhatt et al. [24] reported that enterococcal strains exhibited up to 95% of beta lactum resistance, and this high-level resistance may be due to a low affinity between the penicillin-binding protein of the enterococcal isolates and an antibiotic [28]. We also found that 4% of *E. faecalis* and 11% of *E. faecium* were resistant to linezolid. This same trend of findings was demonstrated in Europe [29], the United States and Taiwan [30], and India [18]. Linezolid resistance among enterococcal isolates remains uncommon in India; however, the current study report indicates a gradual emergence of linezolid resistance in recent years.

The current study’s results also reveal the prevalence rate of MDR (28%), which is defined by resistance to at least three antibiotic categories [31]. The MDR was more frequent among *E. faecium* (52%) than *E. faecalis* (27%), and a similar finding was reported by Maschieto et al. [32]; Karna et al. [1]. The highest proportion of *E. faecium* may be due to its ability to become resistant to multiple antibiotics. Conversely, the current MDR prevalence rate is less than other studies reported from Nepal [1], Ethiopia [23], and Slovenia [25], indicating the lower frequency of MDR in our country. A high-level prevalence rate of MDR isolates was reported from surgical wound infections and blood cultures of patients with bacteremia collected from Minia University hospital, Egypt. The isolates were resistant to cefepime, ampicillin, tetracycline, erytromycin, vancomycin, and linezolid [33]. The prevalence rate of MDR in the current study at least to five antibiotics is about 11% (*n* = 91) and is most frequent among *E. faecium* strains (*n* = 54) compared to *E. faecalis* (*n* = 37), indicating the dissemination of MDR pathogens in healthcare communities. The majority of the vancomycin-resistant enterococcal isolates obtained from the inpatients of Algerian hospitals were found to be resistant to at least five antibiotics in addition to glycopeptides [34].

The minimum inhibitory (MIC) concentration of 5 antibiotics for the 91 VRE isolates (showing resistance to five antibiotics) was determined by the VITEK 2 and MDT assay, and the results are compared in order to discover any discrepancy between the two methods. The study results showed that the MIC values obtained for five antibiotics were higher than the MIC values of previous studies [1,35], indicating the high-level resistance of VRE isolates from Tamil Nadu, a south Indian state. Further, variation in MIC and the sensitivity/resistance pattern of VRE isolates from various countries may be due to sampling size, duration of hospital exposure/stays, geographical location, etc. [1]. The MIC of benzylpenicillin and erythromycin for 91 VRE isolates obtained from the MDT is agreed with their MIC obtained from VITEK 2. However, there was no discrepancy between the reference method (MDT) and VITEK 2 for benzylpenicillin, as well as erythromycin and vancomycin, whereas six minor errors for ciprofloxacin were observed. In line with teicoplanin, one major error and five minor errors were determined between the two methods. These results disclose the reasonable accuracy of the VITEK 2 system. Generally, the VITEK 2 system is an easy system to handle and provides results rapidly with a reasonable accuracy in clinical microbiology laboratories [36].

The pathogenicity of the clinical enterococcal isolates is mainly associated with their virulence factors that facilitate adherence, invasiveness, etc. [37]. In this study, the prevalence of virulence factors among *E. faecalis* and *E. faecium* differed significantly. Further, the frequency of virulence factors such as gelatinase, protease, and hemolysin were higher among *E. faecalis* (73%) compared to *E. faecium* (54%). This result is concurrent with reports from previous studies [13,17,22,25]. Phenotypic studies have also showed that gelatinase and hemolysin are the virulence determinants present with a high frequency among VRE isolates, followed by their biofilm-producing ability. This result is consistent with genotypic studies, in which *gelE* prevalence among VRE isolates is higher than any other genes, followed by *asaI* gene. This result is agreed with by Heidari et al. [38] and Shokoohizadeh et al. [22], who reported *gelE* and *asaI* as the most common virulence genes among VR *E. faecalis*, but is in contrast with Jahansepas et al. [17] who found that *esp* is the most frequent gene present in VRE. The lowest frequency of the gene amongst VRE was *hyl*, and this report is in agreement with the results reported by Jahansepas et al. [17] and Golob et al. [25]). The *cylA* gene was detected in 62% of *E. faecalis* and 56% of *E. faecium* strains, and this is in contrast with Jahansepas et al. [17] who did not detect the *cylA* gene in any of the 35 VRE isolates. A simultaneous presence of all the seven virulence genes was observed in 9% of VRE isolates (*n* = 8, three from *E. faecalis* and five from *E. faecium*).

The study also revealed that the majority of the isolates showing themselves as phenotypically positive for hemolysis, confirmed genotypically via expressing *cylA* gene as hemolysis, is explained by *cylA* gene, which is a prerequisite for activating *cylL*_L/S_ [39]. Few isolates positive for hemolysis due to a lack of *cylA* gene may be explained by a low level or the downregulation of gene expression. Overall, a good correlation was observed between gelatinase phenotype and its biomarker gene *gelE*, protease phenotype and *sprE* gene, and hemolysis phenotype and *cylA* gene. However, the biofilm-forming phenotype was majorly associated with *esp*, *ace*, and *asaI* genes. The virulence genes associated with the biofilm-forming ability vary from enterococcal isolates.

The distribution of virulence genes is highly frequent among *E. faecalis* compared to *E. faecium* isolates. A high frequency of multiple determinants could effectively contribute to bacterial colonization and their pathogenesis among humans. The majority of the isolates (more than 85%) in this study harbored the *vanA* gene and showed high-level resistance to vancomycin (MIC > 32 µg/mL) and teicoplanin (MIC > 32 µg/mL), which is the critical feature of the VanA phenotype. This result is correlated with Ghoshal et al. [40] in India, Talebi et al. [41] in Iran, and Benamrouche et al. [34] in Algeria, who reported the presence of *vanA* gene and the complete absence of *vanB* among the VRE isolates. Recent studies by George et al. [42] in Southi Arabia and Moosavian et al. [43] in Iran reported that the majority of vancomycin resistance (more than 60%) among enterococcal isolates is due to *vanA* gene; nearly less than 5% of the isolates harbored *vanC* gene, and the remaining isolates showed themselves to be positive to *vanB* gene. The major reason for the differences in the existence of resistance among the various regions is mostly based on the usage of antibiotics in hospitals [17].

## 4. Materials and Methods

### 4.1. Study Design and Period of Sampling

The study was carried out at various tertiary care hospitals located in and around Tamil Nadu, a south Indian state, from 2018 to 2020. Specimens including urine, blood, pus, tissue, catheter, vaginal wash, semen, and endotracheal wash were collected from both outpatients and inpatients, including those on ICU, surgery, pediatric, urology, and hematology wards. Samples were collected from every patient after obtaining their written consent in a well-structured questionnaire to collect the data of the study participants. The patients’ historical and clinical data were obtained by 2 experienced nurses.

All the media, reagents, and antibiotics used in this study were purchased from Hi media, India.

### 4.2. Screening and Identification of Enterococcus

All the collected specimens (urine, pus, tissue, catheter, vaginal wash, semen, and endotracheal wash) were inoculated onto blood agar, chocolate agar, and Mac Conkey agar using the spread plate method. All the plates were incubated at 37 °C for 48 h. The blood samples were inoculated into brain heart infusion broth and incubated at 37 °C for 24 h. The pure culture of each isolate was obtained by subculturing them 3–4 times. Enterococcal isolates were screened using colony morphology, Gram’s staining, and bile esculin hydrolysis. Among the enterococcal isolates, *Enterococcus faecalis* and *Enterococcus faecium* were differentiated based on mannitol, sorbitol, arabinose fermentation, and arginine hydrolysis. Similarly, *Enterococcus hirae* and *Enterococcus durans* were differentiated based on acid production from melibiose, and sucrose as *E. hirae* are able to produce acid from melibiose and sucrose, whereas *E. durans* is negative in the fermentation of both sugars [44].

### 4.3. Antimicrobial Susceptibility Test (AST)

All the identified enterococcal isolates were subjected to an AST using the antibiotic disk diffusion technique (ADT) by following the CLSI guidelines (2017). The bacterial isolates were grown in trypticase soy broth and diluted in saline to reach 0.5 McFarland standards. The AST was performed on Muller Hinton agar (MHA) against benzylpenicillin (10 units), high-level gentamycin (synergy) (120 µg), ciprofloxacin (5 µg), levofloxacin, erythromycin (15 µg), linezolid (30 µg), teicoplanin (30 µg), vancomycin (30 µg), tetracycline, tigecycline (15 µg), and nitrofurantoin (300 µg). Vancomycin-resistant enterococci (VRE) were preliminarily screened by ADT. Further, the minimum inhibitory concentration (MIC) of 5 antibiotics (benzylpenicillin, erythromycin, ciprofloxacin, teicoplanin, and vancomycin) was determined using the microtube dilution technique (MDT) (as the reference method) and an AST P628 VITEK-2 system (BioMerieux^TM^, Cambridge, MA, USA) by following the CLSI guidelines (2017) and the manufacturer’s instructions, respectively.

### 4.4. Phenotypic Analysis of Virulence Factors among VRE

The VRE isolates were phenotypically assessed for their virulence factors, such as gelatinase, protease, hemolysis activity, and biofilm production. The positive control (methicillin-resistant Staphylococcus aureus: gelatinase, protease, and biofilm production, and β-hemolytic Streptococcus culture: hemolysis) for each test was used to compare the VRE in all the tests.

#### 4.4.1. Gelatinase Activity

Gelatinase assay was performed by inoculating the isolates in brain heart infusion (BHI) agar supplemented with gelatin (3%, Sigma, Bangalore, India) and incubated at 37 °C. After 48 h of incubation, Frazier solution was flooded on the plate, and a halo zone was observed around the colony.

#### 4.4.2. Protease Activity

The protease-producing ability of the isolates was assessed by inoculating the isolates in BHI agar supplemented with 1.5% of skimmed milk and incubated at 37 °C for 48 h. The clear halo zone around the colonies indicates the protease activity of the isolates.

#### 4.4.3. Hemolytic Activity

The ability of bacterial isolates to lyse the blood cells was analyzed by inoculating the isolates on a blood agar plate (5% defibrinated blood) and incubating at 37 °C up to 72 h. The clear halo zone around the colony indicates the hemolytic potential of the isolates.

#### 4.4.4. Biofilm Production Test

Biofilm assay was performed using the quantitative microtiter plate method as described by Stepanović et al. [45]. Briefly, the colonies of each isolate were suspended in physiological saline to reach the OD_570_ 0.5 McFarland. The wells of the 96-microtiter plate were provided with the addition of 180 µL of trypticase soy broth (TSB) supplemented with 0.5% glucose and 20 µL of the bacterial suspension. The negative control was maintained simultaneously without bacterial suspension. The plate was incubated statistically at 37 °C for 48 h. Then, the broth was drawn off, and three successive washings of the wells were conducted with 300 µL of sterilized phosphate-buffered saline (PBS). The fixative e agent, methanol (200 µL), was added for 20 min, tapped and air dried for 30 min in an inverted position. Finally, the crystal violet stain (200 µL) was added to the biofilm. After 15 min of incubation, the wells were rinsed with water and air dried for 30 min. The wells were provided with destainer, 33% acetic acid (200 µL), and the plate was incubated at room temperature for 30 min under static conditions. The biofilm formation was quantified by measuring the OD_570_ using a microtiter plate reader. The tested isolates were classified based on their biofilm-forming potential according to the OD reading as conducted by Stepanović et al. [45] and Cui et al. [46].

#### 4.4.5. Genotypic Characterization of Virulence Factors from VRE

The isolates showing multi-drug resistance, including vancomycin resistance, were further analyzed for the presence of virulence genes, such as *gelE*, *sprE*, *esp*, *ace*, *asaI*, and *hyl*, *cylA*. In addition, they were assessed for their type of vancomycin resistance using *vanA* and *vanB* primer. Genomic DNA was extracted from the test isolates using QIAamp DNA kits (QIAGEN, Hilden, German). PCR amplification was carried out using various primers, as listed in Appendix A. The PCR products were confirmed by gel electrophoresis (1.2% agarose) and visualized under UV light.

### 4.5. Statistical Analysis

The values are given as percentages. A Chi-square test and Fisher’s exact test were used to correlate the phenotypic and genotypic assessment of the virulence factors of vancomycin-resistant isolates using Graphpad Prism (Version 6). A *p* value < 0.05 was considered statistically significant.

## 5. Conclusions

The results of the current study show that the detection of enterococci from clinical samples is highly imperative. The overall prevalence of VRE and MDR in our study is 31 and 11%, respectively, explaining the high incidence of VRE and MDR strains in the hospital settings. Thus, a regular investigation is a prerequisite in order to prevent the dissemination of MDR pathogens. In addition, the presence of vancomycin-resistant and MDR *E. faecalis* and *E. faecium* strains could be an alarming situation in hospital settings. The high prevalence rate of virulence determinants reported in this study could reveal the necessity of a multifaceted approach to control and prevent the dissemination of MDR pathogens. These results could be of interest for epidemiologists who are engaged in the surveillance of antibiotics of resistance and/or the control of a correct antibiotic policy in the studied geographical area.

## Figures and Tables

**Figure 1 antibiotics-12-00981-f001:**
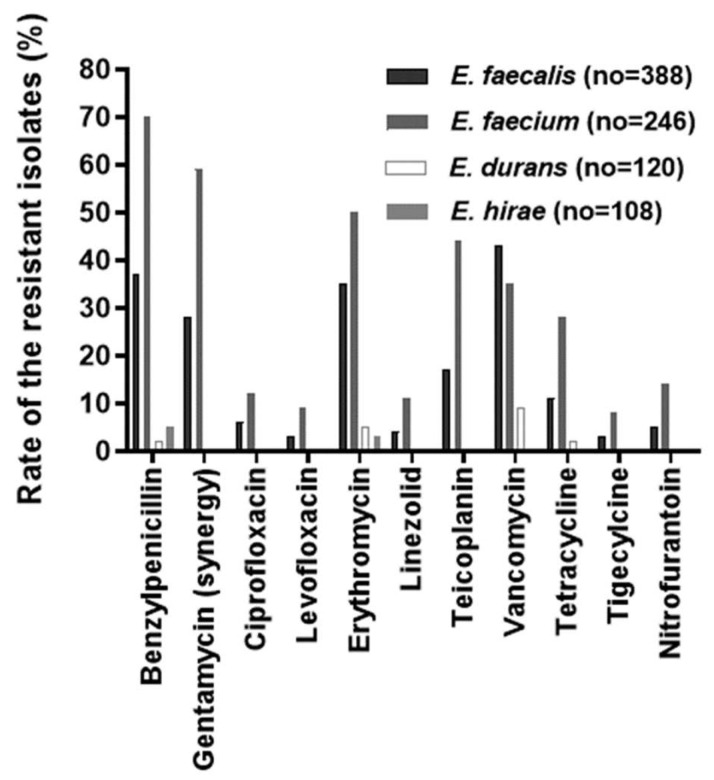
Antibiotic susceptibility/resistance profile of the enterococcal isolates from various clinical specimens determined assessed by antibiotic disk diffusion test (ADT).

**Figure 2 antibiotics-12-00981-f002:**
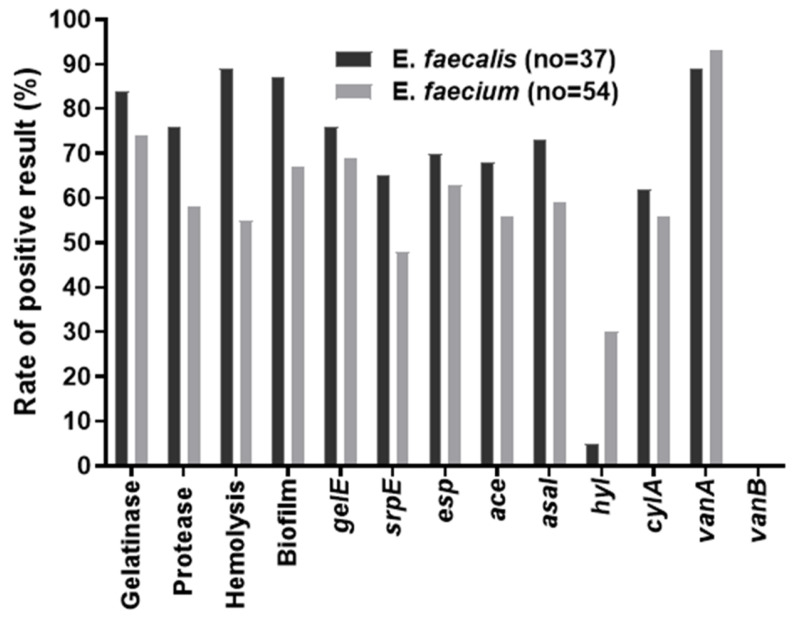
The bar plot showing the percentage of positive phenotypic and genotypic virulence factors of vancomycin-resistant enterococcal isolates.

**Figure 3 antibiotics-12-00981-f003:**
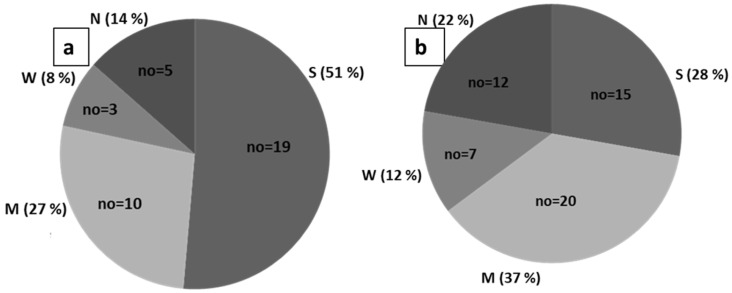
Pie chart revealing the different strength of the biofilm-producing ability of vancomycin-resistant enterococcal isolates (**a**) *E. faecalis* (*n* = 37), (**b**) *E. faecium* (*n* = 54). S: strong, M: moderate, W: weak, N: negative.

**Table 1 antibiotics-12-00981-t001:** Various specimens collected from different age groups of patients.

Type of Specimen	Age Group in Years										Total		
	0–10		11–20		21–40		41–60		61–80				Total
	M	F	M	F	M	F	M	F	M	F	M	F	
Urine	18	21	178	221	120	284	74	143	41	21	431	690	1121
Blood	11	13	31	24	48	23	27	25	15	17	132	102	234
Pus	5	7	13	9	17	13	13	3	6	0	54	32	86
Tissue	3	2	8	5	5	3	2	0	0	0	18	10	28
Catheter	0	0	0	0	2	0	4	0	0	0	6	0	6
Vaginal wash	0	0	0	2	0	14	0	16	0	0	0	32	32
Semen	0	0	27	0	21	0	0	0	0	0	48	0	48
Endotracheal wash	0	0	0	4	3	1	5	1	2	4	10	10	20
Total											699	876	1575

**Table 2 antibiotics-12-00981-t002:** Socio-demographic analysis of samples collected in this study.

Socio-Demographic Details	Frequency (No/%)				
	*Enterococcus faecalis* (388/45%)	*Enterococcus faecium* (246/29%)	*Enterococcus durans* (120/14%)	*Enterococcus hirae* (108/13%)	Total (862/100%)
Sex					
Male	166/43%	72/29%	47/39%	63/58%	348/40%
Female	222/57%	174/71%	73/61%	45/42%	514/60%
Age					
1–5	28/7%	14/6%	0	4/4%	46/5%
6–20	39/10%	19/8%	8/7%	11/10%	77/9%
21–40	68/18%	17/7%	54/45	39/36%	178/21%
41–65	167/43%	121/50%	37/31%	34/32%	359/42%
>65	86/22%	75/31%	21/18%	20/19%	202/23%
Nature of native					
Rural	126/33%	33/13%	46/38%	37/34%	242/28%
Urban	262/68%	213/87%	74/62%	71/66%	620/72%
Patients visited ward					
Out patient	174/45%	55/22%	20/17%	32/30%	281/33%
ICU	24/6%	19/7.7%	-	-	43/5%
Surgery	20/5%	15/6%	10/8.3%	-	45/5%
Pediatric	14/4%	10/4%	-	-	24/3%
Urology	111/29%	108/44%	62/52%	48/44%	329/38%
Hematology	45/12%	39/16%	28/23%	28/26%	140/16%
Specimen collected					
Urine	164/42%	98/40%	47/39%	26/24%	335/39%
Blood	36/9%	24/10%	19/16%	16/15%	95/11%
Pus	60/16%	35/14%	28/23%	21/19%	144/17%
Tissue	27/9%	28/11%	11/9%	13/12%	79/9%
Catheter sample	62/16%	33/13%	10/8%	18/17%	123/14%
Vaginal wash	28/7%	21/9%	4/3%	10/9%	63/7%
Semen	4/1%	7/3%	0	0	11/1%
Endotracheal secretion	7/2%	0/0	1/1%	4/4%	12/1%

**Table 3 antibiotics-12-00981-t003:** Prevalence of multi-drug-resistant phenotype of the enterococcal isolates determined by ADT.

*Enterococcal* sp.	Resistant to 3 Antibiotics	Resistant to 4 Antibiotics	Resistant to 5 Antibiotics
*E. faecalis*	106/27.3	72/18.6	37/9.6
*E. faecium*	129/52.4	91/37	54/22
*E. durans*	5/4.2	0	0
*E. hirae*	0	0	0
Total	240/27.8	163/18.9	91/10.6

**Table 4 antibiotics-12-00981-t004:** MIC of the tested antibiotics (benzyl penicillin, erythromycin, ciprofloxacin, teicoplanin, and vancomycin) for 91 VRE isolates determined by VITEK 2 system and micro tube dilution (MDT) assay.

Antibiotics	MIC Value	S/I/R Pattern	VITEK 2 System *	MDT *
Benzylpenicillin	<1	S	0	0
	2		0	0
	4		0	0
	8	I	0	0
	16	R	19	19
	>32		72	72
Erythromycin	0.12	S	0	0
	0.25		0	0
	0.5		0	0
	1.0	I	0	0
	2		0	0
	4	R	34	34
	>8		57	57
Ciprofloxacin	<1	S	0	0
	2		0	0
	4	I	0	7
	8	R	91	84
	16		0	0
	>32		0	0
Teicoplanin	<1	S	0	0
	2		0	3
	4	I	5	0
	8		0	0
	16	R	6	0
	>32		80	88
Vancomycin	<1	S	0	0
	2		0	0
	4		0	0
	8		0	0
	16	I	0	0
	>32	S	91	91

* According to Chi square test the MIC value determined by VITEK system and MDT was statistically significant (*p* < 0.006).

**Table 5 antibiotics-12-00981-t005:** Discrepancies between VITEK 2 system and the reference method, MDT, in determining the minimum inhibitory concentration of antibiotics against 91 enterococcal isolates.

Antibiotics	No. of Errors		
	Very Major	Major	Minor
Benzyl penicillin	0	0	0
Erythromycin	0	0	0
Ciprofloxacin	0	0	6
Teichoplanin	0	1	5
Vancomycin	0	0	0

**Table 6 antibiotics-12-00981-t006:** Phenotypic and genotypic correlation between gelatinase activity and *gelE* gene.

Genotype	*E. faecalis* (*n* = 37)			*E. faecium* (*n* = 54)		
Phenotype	*gelE* + ve (*n*)	*gelE* + ve (%)	Total	*gelE* + ve (*n*)	*gelE* + ve (%)	Total
Gelatinase + ve	28	90	31	37	88	42
Gelatinase − ve	0	0	6	0	0	12
Total	28	90	37	37	88	54
	Chi-square *p* < 0.0001			Chi-square *p* < 0.0001		
	Fisher’s exact test *p* < 0.0001			Fisher’s exact test *p* < 0.0001		

**Table 7 antibiotics-12-00981-t007:** Phenotypic and genotypic correlation between protease activity and *sprE* gene.

Genotype	*E. faecalis* (*n* = 37)			*E. faecium* (*n* = 54)		
Phenotype	*sprE* +ve (*n*)	*sprE* +ve (%)	Total	*sprE* +ve (*n*)	*sprE* +ve (%)	Total
Protease + ve	16	73	22	26	84	31
Protease − ve	8	13	15	0	100	23
Total	24	65	37	26	48	54
	Chi-square *p* = 0.03884			Chi-square *p* < 0.0001		
	Fisher’s exact test *p* = 0.03003			Fisher’s exact test *p* < 0.0001		

**Table 8 antibiotics-12-00981-t008:** Phenotypic and genotypic correlation between hemolysis activity and *cylA* gene.

Genotype	*E. faecalis* (*n* = 37)			*E. faecium* (*n* = 54)		
Phenotype	*cylA* + ve (*n*)	*cylA* + ve (%)	Total	*cylA* + ve	*cylA* + ve (%)	Total
Hemolysis + ve	23	70	33	28	93	30
Hemolysis − ve	0	100	4	2	8	24
Total	23	62	37	30	56	54
	Chi-square *p* < 0.0001			Chi-square *p* = 0.0301		
	Fisher’s exact test *p* < 0.0001			Fisher’s exact test *p* = 0.0152		

**Table 9 antibiotics-12-00981-t009:** Phenotypic and genotypic correlation between biofilm-forming ability and *gelE* and *sprE* genes.

Genotype	*E. faecalis* (*n* = 37)				*E. faecium* (*n* = 54)			
Phenotype	gelE + ve (*n*)	gelE + ve (%)	sprE + ve (*n*)	sprE + ve (%)	gelE + ve (*n*)	gelE + ve (%)	sprE + ve (*n*)	sprE + ve (%)
Strong	17	46	15	41	11	20	9	17
Medium	8	22	8	22	11	20	8	15
Weak	2	5	0	0	7	13	3	6
Biofilm negative	1	3	1	3	8	15	6	11
	Chi-square *p* = 0.9388		Chi-square *p* = 0.02		Chi-square *p* = 0.3595		Chi-square *p* = 0.7417	
	Fisher’s exact test *p* = 0.3996		Fisher’s exact test *p* = 0.01		Fisher’s exact test *p* = 0.3029		Fisher’s exact test *p* = 0.6750	

**Table 10 antibiotics-12-00981-t010:** Phenotypic and genotypic correlation between biofilm-forming ability and *esp, ace*, and *asaI* genes.

Genotype	*E. faecalis* (*n* = 37)						*E. faecium* (*n* = 54)					
Phenotype	*esp* + ve (*n*)	*esp* + ve (%)	*ace* + ve (*n*)	*ace* + ve (%)	*asaI* + ve (*n*)	*asaI* + ve (%)	*esp* + ve (*n*)	*esp* + ve (%)	*ace* + ve (*n*)	*ace* + ve (%)	*asaI* + ve (*n*)	*asaI* + ve (%)
Strong	15	41	17	46	17	46	15	28	14	1	1	2
Medium	8	22	7	19	9	24	14	26	13	1	13	24
Weak	0	0	1	3	1	3	5	2	4	4	4	7
Biofilm negative	1	3	0	0	0	0	0	18	33	18	1	2
	Chi-square *p* = 0.02		Chi-square *p* = 0.0215		Chi-square *p* = 0.0450		Chi-square *p* = 0.0411		Chi-square *p* = 0.0084		Chi-square *p* = 0.0428	
	Fisher’s exact test *p* = 0.01		Fisher’s exact test *p* = 0.0129		Fisher’s exact test *p* = 0.0365		Fisher’s exact test *p* = 0.0333		Fisher’s exact test *p* = 0.0076		Fisher’s exact test *p* = 0.0404	

## Data Availability

Not applicable.

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
