# Peer review of "Antibiotic Susceptibility Patterns and Virulence-Associated Factors of Vancomycin-Resistant Enterococcal Isolates from Tertiary Care Hospitals"

_antibiotics, 2023, doi:10.3390/antibiotics12060981_

Round 1
Reviewer 1 Report
Major suggestions
§ Please clearly define the MDR together with reference or previous literature article (up to 3 or 5 antimicrobial drugs or classes).
§ This work needs to perform molecular assays (PCR plus Sequencing or PCR detection of marker gene for each species) to confirm species identification and reference strains should be included for each species.
§ The data presented in Table 2, Table 3, Table 4, and Table 5 need to be compared by biostatistical analysis.
§ Re-check and fit the best statistics for frequency comparisons of two groups and multiple groups (Table 6, 7, 8, 9, and 10). Need to analyze associations between virulence phenotype and patterns of virulence genes as well.
§ Authors need to include control strains of “4.4. Phenotypic analysis of virulence factors among VRE”.
Minor suggestions
§ Line2: correct “Antibiotic Susceptibility Pattern” as plural nouns
§ Line16-33: should add the data of MDR-VRE
§ Line17: correct “no=” and check the whole manuscript
§ Line41-43: add reference(s)
§ Line45-46: add reference(s)
§ Line38-56: should include the epidemiology of VRE in the country studied or related country.
§ LineWholeMS: correct “disc diffusion test (DDT)” and “antibiotic disk diffusion test (ADT)”
§ Line158: How many isolates of VR E. faecalis and E. faecium obtained from out- vs hospitalized patients.
§ Line213-216 : please show the number of isolates along with percentage (Table 6, 7, 8, 9, and 10). Authors already show the data of virulence genes +ve. So, should exclude the data of virulence gene -ve.
§ Line237-251: discuss the rate of drug resistance, specimen, clinical data and pathogenicity of VE isolated from out- vs hospitalized patients
§ Line276-286: please add discussion about the important of MDR-VE and their epidemiology
Moderate editing of English language
Author Response
Dear Reviewer,
Thank you for your valuable suggestions and comments. Surely these corrections will improve the manuscript. The reviewer’s comments are considered meticulously and corrected in the revised manuscript.
Reviewer 1 Comments (Major) and Authors Response
Comment 1: Please clearly define the MDR together with reference or previous literature article (up to 3 or 5 antimicrobial drugs or classes).
Authors Response: Thank you for your suggestion. MDR bacteria are resistant to three or more antimicrobial classes (Magiorakos et al., 2012) (Mentioned in line number 281). In our study, we have used eight different classes of antibiotics to find out the prevalence of multi drug resistant enterococci. According to our study, 106 isolates were found to be resistant to three antimicrobial groups (multi-drug resistant pathogens) such as beta-lactum antibiotics, glycopeptides and macrolides.
Comment 2: This work needs to perform molecular assays (PCR plus Sequencing or PCR detection of marker gene for each species) to confirm species identification and reference strains should be included for each species.
Authors Response: Thank you for your suggestions. In this study we have identified a large number of vancomycin resistant enterococcal strains. It is big task for sequencing all these isolates due to financial shortage as we don’t have much financial assistance from funding agencies. Thus, we selected only 6 strains from 106 isolates and sequenced for 16S rRNA sequenced and submitted that data as a separate paper in Applied Journal of Biochemistry and Biotechnology (under revision). This study mainly focused on vancomycin resistant Enterococcus sp. According to reviewer’s suggestion, it is very difficult to use reference strain for each species as this kind of multi drug resistant strains of Enterococci are not available even in MTCC and ATCC.
Comment 3: The data presented in Table 2, Table 3, Table 4, and Table 5need to be compared by biostatistical analysis.
Authors Response: Thank you for valuable comment to improve the article. Table 2 showed only socio demographic details of the study. So, in this table statistical analysis is not generally performed. Table 3 results are mean values of 3 independent experiments. The SD values have not mentioned as it is 0.0 for all the mean values. In the revised manuscript Table 4 data are statistically analyzed by Chi-Square test and included. Table 5 showed the discrepancy details between the VITEK system and Micro tube dilution method.
Comment 4: Re-check and fit the best statistics for frequency comparisons of two groups and multiple groups (Table 6, 7, 8, 9, and 10). Need to analyze associations between virulence phenotype and patterns of virulence genes as well.
Authors Response: Thank you for your valuable suggestion. In table 6,7,8,9 and 10 Chi square test and Fisher’s exact test were used for frequency comparison of Enterococcus faecalis and E. faecium. The same test was used by Siju Kankalil et al. (2021) and Hashem et al. (2021) for the frequency comparison.
Comment 5: Authors need to include control strains of “4.4. Phenotypic analysis of virulence factors among VRE”.
Authors Response: Thank you for your valuable comment. The positive control strains for all the virulence factors are included in the revised manuscript.
Reviewer 1 Comments (Minor) and Authors Response
Thank you for all the reviewer positive comments and suggestions to improve the manuscript. The comments are taken meticulously and corrected in the revised manuscript.
Comment 1: Line 2: correct “Antibiotic Susceptibility Pattern” as plural nouns
Authors Response: “Antibiotic Susceptibility Pattern” is corrected as “Antibiotic Susceptibility Patterns” in revised manuscript.
Comment 2: Line16-33: should add the data of MDR-VRE
Authors Response: Data of MDR-VRE has added in Line no. 22-25. Now VRE is included in the revised manuscript.
Comment 3: Line17: correct “no=” and check the whole manuscript
Authors Response: ‘no=’ is replaced with ‘n=’ in the revised whole manuscript.
Comment 4: Line 41-43: add reference(s)
Authors Response: Reference is added in the revised manuscript.
Comment 5: Line 45-46: add reference(s)
Authors Response: Reference is added in the revised manuscript.
Comment 6: Line38-56: should include the epidemiology of VRE in the country studied or related country.
Authors Response: As per reviewer’s comment, epidemiology of VRE in India is included in the revised manuscript.
Comment 7: Line Whole MS: correct “disc diffusion test (DDT)” and “antibiotic disk diffusion test (ADT)”
Authors Response: Corrected in revised manuscript.
Comment 8: Line158: How many isolates of VR E. faecalis and E. faecium obtained from out- vs hospitalized patients.
Authors Response: Thank you for the reviewer valuable query. A total of 167 Enterococcus faecalis and 86 Enterococcus faecium from 1565 isolates were found to various clinical specimens were vancomycin resistant.
Further, we have not analyzed the hospitalized and out and patients samples separately. However, the enrolled 1212 were hospitalized samples and 363 were out patients.
Comment 9: Line 213-216: please show the number of isolates along with percentage (Table 6, 7, 8, 9, and 10). Authors already show the data of virulence genes +ve. So, should exclude the data of virulence gene -ve.
Authors Response: Thanks for the valuable comment. Correction is carried out in Tables 6,7,8,9 and10 in revised manuscript.
Comment 10: Line 237-251: discuss the rate of drug resistance, specimen, clinical data and pathogenicity of VE isolated from out- vs hospitalized patients.
Authors Response: Included in the revised manuscript.
Comment 11: Line 276-286: please add discussion about the important of MDR-VE and their epidemiology.
Authors Response: Thank you for your suggestion. The discussion was included in the revised manuscript.
Reviewer 2 Report
The manuscript presents the data characterizing the enterococcal isolates collected from outpatients and inpatients. Medium correction needed for the manuscript before publication.
Comments.
Line 22. “bezylpenicillin” must be corrected on “benzylpenicillin”.
Line 24. “multi-drug resistance (resistant to 5 antibiotics)”
Lines 52-53 “Gupta et al., 2020; Banerjee 52 and Anupurba, 2016; Mathur et al., 2003” should be presented in in chronological order.
Line 81. “cylA” should be replaced on “СylA”.
Lines 91-93 and so on. “no=” should be replaced on “n=”.
Table 2. “Socio-demographic details” should be replaced on “Details”.
Line 112. “(no=) ????
Figure 1. “% of resistance” should be replaced on “Rate of the resistant isolates, %”.
Lines 109, 123. (DDT) and (ADT) abbreviations - one of them should be selected.
Table 3. Bacterial names must be in italics.
Line131. “resistant” should be replaced by “resistance”.
Table 4 should be deleted because the data was already presented in the text.
Table 5 should be deleted. Differences between MIC values of adjacent values are not errors - they are within the accuracy of the method.
Line 164. Correct please “(28 (76 %)”.
Figure 2. “% of positive results” should be replaced on “Rate of positive results, %”.
Figure 3. Are the data presented statistically reliable for such a small sample?
Lines 187-190. The data are duplicated in Table 2.
Tables 6-8 should be combined.
Tables 9-10 should be combined.
Line 226. The “major” should be replaced on “one of the important” or “one of the major”.
Materials and Methods. The section “Bioethical requirements” must be added. Permission from the Ethics Committee to conduct the study must be submitted.
Authors must to present the manufacturers for all reagents and nutrient medias.
References are not presented in accordance with the requirements of the journal.
Reference 34 should be completed.
The text must be carefully checked.
Author Response
Response to Reviewer 2 comments
Dear Reviwer 2,
Thank you for your valuable suggestions and comments. Surely these corrections will improve the manuscript. All the reviewer’s comments are considered meticulously and corrected in the revised manuscript
Reviewer 2 Comments and Authors Response
Comment 1: Line 22. “bezylpenicillin” must be corrected on “benzylpenicillin”.
Authors Response: Corrected in the revised manuscript.
Comment 2: Line 24. “multi-drug resistance (resistant to 5 antibiotics)”
Authors Response: Changed in the revised manuscript.
Comment 3: Lines 52-53 “Gupta et al., 2020; Banerjee 52 and Anupurba, 2016; Mathur et al., 2003” should be presented in chronological order.
Authors Response: Corrected in the revised manuscript.
Comment 4: Line 81. “cylA” should be replaced on “СylA”.
Authors Response: According to reviewer’s comment changed in the revised manuscript.
Comment 5: Lines 91-93 and so on. “no=” should be replaced on “n=”.
Authors Response: Thanks for your correction and corrected in the revised whole manuscript.
Comment 6: Table 2. “Socio-demographic details” should be replaced on“Details”.
Authors Response: Corrected in the revised manuscript.
Comment 7: Line 112. “(no=) ????
Authors Response: The number of isolates is included in the revised manuscript.
Comment 8: Figure 1. “% of resistance” should be replaced on “Rate of the resistant isolates, %”.
Authors Response: Changed in the figure.
Comment 9: Lines 109, 123. (DDT) and (ADT) abbreviations - one of them should be selected.
Authors Response: Corrected in the revised manuscript.
Comment 10: Table 3. Bacterial names must be in italics.
Authors Response: Corrected in the revised manuscript tables.
Comment 11: Line131. “resistant” should be replaced by “resistance”.
Authors Response: Corrected in the revised manuscript.
Comment 12: Table 4 should be deleted because the data was already presented in the text.
Authors Response: The tables easily explain the results without reading the title of manuscript which may helpful for the followers. Thus, please permit us to include the tables.
Comment 13: Table 5 should be deleted. Differences between MIC values of adjacent values are not errors - they are within the accuracy of the method.
Authors Response: Table 5 showed the number of minor and major errors between VITEK and ADT. So, this table proved the efficiency of VITEK system result and that is the reason to keep the table. If table is removed, VITEK result could not be convincible. Kindly permit us to keep the table.
Comment 14: Line 164. Correct please “(28 (76 %)”.
Authors Response: Corrected in the revised manuscript.
Comment 15: Figure 2. “% of positive results” should be replaced on “Rate of positive results, %”.
Authors Response: Corrected in the revised manuscript figure.
Comment 16: Figure 3. Are the data presented statistically reliable for such a small sample?
Authors Response: The biofilm experiment was carried out in triplicate and the mean values only showed in figure and the standard deviation of the experiment was zero, it was not showed in figure.
Comment 17: Lines 187-190. The data are duplicated in Table 2.
Authors Response: All the data showed in table 2 was not given in line 187-190. So, the table explains all the data. Further, if the table is included, it will easily explain the results.
Comment 18: Tables 6-8 should be combined.
Authors Response: If the tables 6-8 will be combined, that becomes very lengthy and as per the reviewer 1 suggestion, we added percentage of all the result for each genotype like Gel, Spr, CylA for E. faecalis and E. faecium.
Comment 19: Tables 9-10 should be combined.
Authors Response: If the tables 6-8 will be combined, that becomes very lengthy and as per the reviewer 1 suggestion, we added percentage of all the result for each genotype like esp, ace, asaI, etc.
Comment 20 Line 226. The “major” should be replaced on “one of the important” or “one of the major”.
Authors Response: Corrected in the revised manuscript.
Comment 21: Materials and Methods. The section “Bioethical requirements” must be added. Permission from the Ethics Committee to conduct the study must be submitted.
Authors Response: Bioethical statement has given at the end of the manuscript before the reference section.
Comment 22: Authors must to present the manufacturers for all reagents and nutrient medias.
Authors Response: Included in the revised manuscript.
Comment 23: References are not presented in accordance with the requirements of the journal.
Authors Response: Corrected in the revised manuscript.
Comment 24: Reference 34 should be completed.
Authors Response: Reference 34 is the book reference and that is the complete details of the book.
Reviewer 3 Report
The manuscript “Antibiotic Susceptibility Pattern and Virulence Associated Factors of Vancomycin Resistant Enterococcal Isolates from Tertiary Care Hospitals” is good piece of work, however there are some major concerns.
The manuscript titled "Antibiotic Susceptibility Pattern and Virulence Associated Factors of Vancomycin Resistant Enterococcal Isolates from Tertiary Care Hospitals" is a well-written piece of work. However, there are a few major concerns that need to be addressed.
The study analyzed 1575 clinical specimens, including urine, blood, pus, tissue, catheter, vaginal wash, semen, and endotracheal secretions, and found 862 enterococcal isolates. This is a high number and prompts the question of whether there was some sort of outbreak.
The authors noted that the vanA gene was present in the majority of the VR enterococcal isolates (88 out of 91), while none of them harbored the vanB gene. What about the remaining 3 isolates? Was the resistance mechanism evaluated?
Additionally, was there any difference in the prevalence of virulence factors among the different types of specimens?
The authors have mentioned “The overall prevalence of VRE and MDR in our study is 31 and 11 % respectively”. How were the isolates categorized as MDR?
The English quality is fine with only minor grammatical errors
Author Response
Authors response to Reviewer 3 comments
Dear Reviewer 3,
Thank you for your valuable suggestions and comments. These corrections will surely improve the manuscript. All the comments and suggestions are considered meticulously and corrected in the revised manuscript.
Reviewer 3:
Comment 1: The study analyzed 1575 clinical specimens, including urine, blood, pus, tissue, catheter, vaginal wash, semen, and endotracheal secretions, and found 862 enterococcal isolates. This is a high number and prompts the question of whether there was some sort of outbreak.
Authors Response: In this study, 1575 clinical specimen were enrolled. From this large number of specimens, nearly 5,675 bacterial strains were isolated. Among them 862 well grown enterococcal strains were screened. We collected specimen from hospitalized and outpatients not specific to particular disease. Further, the study was conducted continuously about 2 years. So, this number is enterococcal isolates is not indicating outbreak like situation.
Comment 2: The authors noted that the vanA gene was present in the majority of the VR enterococcal isolates (88 out of 91), while none of them harbored the vanB gene. What about the remaining 3 isolates? Was the resistance mechanism evaluated?
Authors Response: Among the 91 isolates, 88 were found to be positive for vanA gene. This result was supported by Esmail et al. (2019) and Manson et al (2003) who reported that vanA is dominated among the VRE. The remaining 3 isolates were vancomycin resistant and proved by antibiotic disk diffusion test and VITEK 2 system, but they were negative for vanA gene. The non-vanA VRE isolates might be less resistant to vancomycin and have less resistant genotype. This hypothesis needs to be further studied in our future studies.
Comment 3: Additionally, was there any difference in the prevalence of virulence factors among the different types of specimens?
Authors Response: Our study results showed that high rate of prevalence of virulence factors was observed among the urine sample and followed by outpatient sample. This result was not included in the manuscript as we highly focused on the species of Enterocoocus.
Comment 4: The authors have mentioned “The overall prevalence of VRE and MDR in our study is 31 and 11 % respectively”. How were the isolates categorized as MDR?
Authors Response: The isolates resistant to more than 3 antibiotic categories were considered as MDR according to Magiorakos et al. (2012). In our study, we have used eight different classes of antibiotics to find out the prevalence of multi drug resistant enterococci. According to our study, 106 isolates were found to be resistant to three or more antimicrobial groups (multi-drug resistant pathogens) such as beta-lactum antibiotics, glycopeptides and macrolides, etc.
Round 2
Reviewer 3 Report
Authors Response: In this study, 1575 clinical specimen were enrolled. From this large number of specimens, nearly 5,675 bacterial strains were isolated.
It is peculiar that there will be numerous specimens considered negative, as at least 50 percent of urine samples yield negative results. Similarly, blood cultures tend to be negative in the majority of cases. The presence of 5,675 bacterial strains from 1575 is highly unusual. Furthermore, there was a mixed growth observed ?. It raises the question of whether this mixed growth was deemed significant.
Minor editing of English language required including spell check and syntax